



# Control Spectra for Quito

Roberto Aguiar[1], Alicia Rivas-Medina[1,2], Pablo Caiza[1], Diego Quizanga[3]

[1] Departamento de Ciencias de la Tierra y la Construcción, Universidad de Fuerzas Amadas ESPE, Quito, Ecuador
[2] Departamento de Cs. Geodésicas y Geomática, Universidad de Concepción - Campus Los Ángeles, Los Ángeles, Chile.
[3] Departamento de Ingeniería Civil, Escuela Politécnica Nacional, Quito, Ecuador

*Correspondence to*: Roberto Aguiar (rraguiar@espe.edu.ec)

**Abstract.** The Metropolitan District of Quito is divided into five areas: south, south-central, central, north-central and north. It is located on or very close to segments of reverse blind faults: Puengasí, Ilumbisí-La Bota, Carcelen-El Inca, Bellavista-Catequilla and Tangahuilla as indicated in Alvarado et al., 2014, making it one of the most seismically dangerous cities in

the world. For each of the urban areas of Quito, elastic response spectra are presented in this paper, which are found using some of the new models of the PEER's NGA-West2 Program, models developed by: Abrahamson et al., 2013; Campbell and Bozorgnia, 2013; and Chiou and Youngs, 2013. These spectra are calculated considering the maximum amount that could be generated by the rupture of each fault segments, and taking into account the soil type that exists in each zone according to the Norma Ecuatoriana de la Construcción, 2015 (NEC-15). Subsequently, the recurrence period of earthquakes of high

magnitude in each fault segment is determined from the physical parameters of the fault segments (size of the fault plane and slip rate), and considering that the fault can break in earthquakes of magnitude less than the expected maximum (minimum size 5.0 Mw). For this, the pattern of recurrence of type GR earthquakes (Gutenberg and Richter, 1944) with double truncation magnitude (*Mmin* and *Mmax*) proposed by Cosentino et al., 1977 is used.

## 1 Introduction

Ecuador is located in one of the most seismically dangerous areas of the world, a zone where the Nazca plate subducts under the American plate at a relative speed of 58 mm $\pm$ 2 mm per year (Trenkamp et al., 2002). As a result, this tectonic movement has generated the megafault that begins in the Gulf of Guayaquil and ends in the Bocono fault in Venezuela, as shown in Fig. 1.

This megafault, towards northeast is transcurrent dextral and, towards north is a reverse fault, with an average rate of slip
that varies between 3.0 and 4.5 mm per year. (Winter et al., 1993; Trenkamp et al., 2002). Obviously this movement is not uniform along the fault, so that the blind reverse faults system of Quito, SFQ, which is part of the megafault, has a turnover rate of 2-4 mm per year.

Concern about the seismic hazard of Quito dates back to 1587, when an earthquake of magnitude 6.4 (Beauval et al., 2010) associated with the system of blind faults SFQ, affected the young city established in 1534. Since then, there has not been



other magnitude 6.0 earthquake, which means that there is a significant accumulation of seismic energy that eventually will be released. Therefore, the city must await new strong earthquakes.

In Earthquake Engineering, earthquakes are expressed by elastic design spectra. It is also well known that local soil conditions are a key factor affecting the spectrum form. A recent case is the earthquake at Christchurch, New Zealand in

2011, where this earthquake, magnitude 6.2 and focal depth of 5 km, generated different spectra in the commercial area of the city, depending on the soil type, affecting it severely (Elwood, 2013).

In this paper, elastic response spectra, obtained deterministically for 5% damping, are presented for each of the five areas of Quito: South-Central, Central, North and North-Central, and taking into account the soil type. These spectra are associated with the occurrence of an earthquake of maximum expected magnitude in fault segments: Puengasí, Ilumbisis-La Bota;

Carcelen El Inca; Bellavista-Catequilla and Tangahuilla.

Being "similar" the maximum expected magnitude in each fault segment, higher spectral ordinates are obtained due to the fault that is closest to the study area. The spectra were obtained with the strong movement models developed by Campbell and Bozorgnia, 2013 (CB13); Abrahamson et al., 2013 (ASK13) and Chiou and Youngs, 2013 (CY13).

The spectra results for 50% and 84% confidence levels are presented and compared to the spectrum reported by the Norma

Ecuatoriana de la Construcción, 2015 (NEC-15). These spectra are called control spectra since they are used to verify the performance of existing structures or those being designed using the NEC-15 norm.

## 2 Blind faults in Quito

Quito is a south to north elongated city which borders west with the western mountains WC, as shown in Fig. 2. The east is limited by several hills which are: Puengasí (P), Ilumbisí La Bota (ILB), towards north the Inca Calderon hill (CEI), and at

the north the Bellavista Catequilla hill (BC), where the last moderately strong earthquake of August 12, 2014 which had a magnitude of 5.1 and focal depth of 5 km, (Aguiar et al., 2014) originated. It can be observed also the Guayllabamba basin, GB. On the right of Fig. 2, the inter-depression ID is observed, too; it is formed by the Chillos valley at the north, and Tumbaco valley at the south. They are separated by the Ilalo volcano IV.

Quito is 2800 m above the sea level and the valleys 2400 m. The hills: P, ILB, CEI, and BC, have a height that varies

between 100 and 300 m. They are the external signs of the existence of blind reverse faults that raise from 2 to 4 mm, per year. They are shown in Fig. 3. It can be observed that in some areas  houses have been built on both sides of the hills (Puengasí).

Note in Fig. 3 that the top of Puengasí and Ilumbisí-La Bota hills, which shows the existence of inverse faults behind them, do not form a single continuous  line, demonstrating the probable existence of a strike-slip fault between them. All this

makes clear that the city stands on a complex fault system.

In Fig. 3, to the left side of Ilumbisí-La Bota hills, the Simon Bolivar Avenue can be seen, as well as the start of the Tumbaco valley and further south the Chillos valley. On the bottom right side of this figure, at the slopes of the hills, the



Guápulo Shrine is located and, towards north, the Metropolitan Park, where a seismic refraction study was conducted to determine the velocity of shear wave *Vs30* on a rocky outcrop.

Table 1 shows the segments of the thrust faults that cross the city of Quito; the length of the rupture surface was estimated by Alvarado et al., 2014 and the area of rupture and maximum expected magnitude was obtained using equations Leonard, 2010. The average dip angle of the thrust faults is 550 westward.

Even though a fault could be considered as well known, there is always uncertainty in the determination of the parameters defining the geometry of it, which is why strong-motion models should be used that somehow minimize this uncertainty. For example, in the model of CY13; the $Z_{TOR}$ variable is the depth to the upper edge of the fault, and it is worked as an increment. Indeed, $\Delta Z_{TOR}$ is defined as the difference between the observed value and the expected value $E[Z_{TOR}]$.

$$\Delta Z_{TOR} = Z_{TOR} - E[Z_{TOR}] \qquad (1)$$

where $Z_{TOR}$ is the observed depth to the upper edge of a given fault; $E[Z_{TOR}]$ is the average obtained with the Eq. (2), that has been inferred for reverse and oblique reverse faults and with the Eq. (3) for normal and transform faults.

$$Z_{TOR} = \max[2.704 - 1.226 \max(M - 5.849, 0), 0]^2 \qquad (2)$$

$$Z_{TOR} = \max[2.673 - 1.136 \max(M - 4.970, 0), 0]^2 \qquad (3)$$

where $M$ is the expected magnitude in the geological fault and $Z_{TOR}$ is in kilometers. But not only that, it is necessary to work with various models of strong movements that respond to the state of knowledge and worldwide databases. In this work, three comprehensive models have been selected, developed by CB13, ASK13 and CY13.

These models will control the level of confidence with the addition of the $\varepsilon$ and $\sigma$ variables, where $\sigma$ is the standard deviation, and $\varepsilon$ is a number that can be worth 0, so that the confidence level is 50% (average), $\varepsilon = 1$ for a confidence level of 84%, and $\varepsilon = 2$ for a confidence level of 95%. Obviously, the higher the value of ε, the higher the confidence level and the greater the spectral ordinates. This issue, the level of confidence in the determination of the spectral coordinates, could induce higher construction costs. Therefore, seismic regulations allow a probability of exceedance of ground motion as a function of the use of the structure.

## 3 Strong movement models

In this paper the following models of strong movements are considered: Campbell and Bozorgnia, 2013 (CB13), Abrahamson et al., 2013 (ASK13) and Chiou and Youngs, 2013 (CY13), so that 5% damping elastic response spectra for surface type geological faults can be obtained. The database of the first three models is the NGA-West2 PEER, which contains over 21,000 accelerograms for the three components of ground motion of earthquakes recorded in different parts of the world, with magnitudes ranging between 3 and 7.9. From this record wealth, CB13 selected 15521 records from 322 earthquakes, while ASK13 worked with 15749 records of 326 earthquakes. Finally, the model of CY13 worked with 12444



records of 300 earthquakes. From this grand total, 2587 records correspond earthquakes to unregistered in California but in other parts of the world.

All three models have very important databases that accredited their equations of ground motion. Table 2 shows the variables that each of the models considers. It shows that there is little difference between them, in general, so that the spectral shapes tend to be similar.

The CY13 model considers the effect of directivity, which is very important when the site of interest is very close to the fault, unlike the other two models that do not consider it. Its formulation is based on studies by Spudich and Chiou, 2008 and developed by Spudich et al., 2013. In the first study, the IDP directivity factor (Isochrone Directivity Predictor) is considered. Instead, in the second model the directivity parameter DPP is used, but the variables are expressed incrementally, for example $\Delta_{DPP}$ and $\Delta Z_{TOR}$, which are also calculated incrementally in the CY13 model.

 In addition, the CB13 considers the hypocenter depth $Z_{HYP}$ as a function involved in determining the spectral acceleration, variable that is not considered in the other models.

To check any difference, it is important to consider some strong movement models for the determination of the spectra (for case studies) or ground motion attenuation equations (for seismic hazard studies).

## 4 Soil classification in Quito

In Fig. 4, the five zones of the Metropolitan District of Quito are presented: South, South-Central, Central, North-Central and North; for each zone it is necessary to determine response spectra associated with blind reverse faults segments. For example: the Puengasí fault starts in the south and reaches the center of the city, its rupture length, associated with a 6.4 magnitude quake, is 22 km. (Table 1), so that this fault is the source of the highest spectral accelerations in the following areas: north-central, central, south-central and south.

At the north of the city, the fault segment associated with the highest spectral accelerations is Ilumbisi-La Bota (ILB), where an earthquake of 6.2 maximum magnitude, associated with a failure length of 15 km, is expected. The other faults that exist in the north of the city do not generate spectra with higher ordinates due to their longer distances.

In Quito, there have been several studies to classify soils, the first of them presented by the Politécnica Nacional in 1994 from a geological point of view; the second study was presented by Valverde et al., 2002; which classified soils according to the Ecuadorian code of year 2000 (CEC-2000); they identified three types of soils called S1, S2 and S3 (Aguiar, 2003).

In the soil profile S1, the speed of the shear wave is $Vs30 \geq 750$ m/s, and the periods of soil vibration are less than 0.2 sec.; in the soil profile S2, the periods of soil vibration are between 0.2 sec. and 0.6 sec. Finally, in the soil profile S3 the periods are greater than 0.6 sec. This is as a brief summary of the soil types in CEC-2000 so that they can be compared with other soil classifications.

Now, in the Ecuadorian code of year 2015 (NEC-15), there are 6 types of soil. The soil profile A corresponds to competent rock with $Vs30 > 1500$ m/s, the soil profile B corresponds to a mean stiffness rock with $760$ m/s $< Vs30 < 1500$ m/s, the soil





profile C has 360 m/s < *Vs30* < 760 m/s, in the soil profile D, the speed of the shear wave Vs30is between 180 m/s and 360 m/s; in soil profile E the speed is less than 180 m/s. Finally the soil profile F is a very low resistance soil which requires the presence of a specialist in soils or geotechnical engineer to make an assessment.

To get a better idea of Quito soil types, a seismic refraction study was done in the Metropolitan Park, in a place where there is a rocky outcrop about 30 meters high, see Fig. 5. It was found that *Vs30* = 466.27 m/s, so that it is a floor type C (Castillo, 2014).

Definitely, in Quito there is no type A soil, but soil types B, C and D, as shown in Fig. 4. Additionally, it should be remembered that soil types B and C have a very good compressive strength

## 5 New spectra

Three meshes were formed with equidistant points, each one hundred meters apart in the NS and EW directions, and for soil types B, C and D. In turn, these meshes were divided into the five areas considered in the Metropolitan District of Quito. In each area, and for each type of soil, spectra was obtained at each point of the mesh, using the three strong motion models indicated above, and finally, an average spectrum was obtained from these spectra.

In the north area of Quito, an earthquake of magnitude 6.2, related to the fault segment Ilumbisí La Bota, generates the largest spectral accelerations since it is closer to this area. In the first row of Fig. 6, these spectra are indicated for confidence levels of 50% and 84%, and for soils B, C and D. Fault segments Puengasí, Carcelén-El Inca, Bellavista-Catequilla and Tangahuilla generate smaller amplitudes spectra for this area.

In the last four rows of Fig. 6, the spectra for the North-Central, Central, South-Central, and South zones of Quito are presented for an earthquake of magnitude 6.4 on the fault segment of Puengasí. The other fault segments generate lower spectral ordinates

## 6 Recurrence periods

The return period is the time between the occurrence of two events at the same seismic source. Therefore, it is a concept that helps to estimate the expected time of occurrence of an earthquake of a given magnitude.

To estimate the recurrence interval associated with different magnitudes within each segment, the seismic potential of the fault should be estimated first, modeled by the rate of seismic moment. This parameter estimates the annual accumulation of energy in each segment of the fault and will be used to relate the slip rate with the assigned recurrence model.





## 1.1 Step 1. Estimation of the seismic moment rate ($\dot{M}o$) in each fault segment.

From the size of the fault plane of each segment (Table 1) , the rate of the segment slippage (3.0 - 4.0 mm / year by Alvarado et al. 2014), and with the conservative assumption that all the plane fault is accumulating energy evenly, the moment rate $\dot{M}o$ can be related to the above parameters according to expression of Brune, 1968, Eq. (4).

$$\dot{M}_0 = \mu \cdot \dot{u} \cdot A \tag{4}$$

where μ is the shear modulus, $\dot{u}$ the slippage rate and A the fault plane area.

## 1.2 Step 2. Seismicity rate estimation using the modified recurrence model GR and $\dot{M}o$

Recurrence models define the seismic potential failure relating the frequency and size of earthquakes occurring in a particular source in a given time. The parameters used to define the potential seismic quakes are: the number of earthquakes

of a certain magnitude $\dot{n}(m)$ (the inverse of the period of recurrence in a given time unit), or the cumulative rate of earthquakes of a magnitude less than a given value $\dot{N}(m)$, and the proportion of large versus small earthquakes [*b* o *β*].

Depending on the relationship established between these parameters, the literature offers different models Gutenberg and Richter, 1944; Main and Burton, 1981; Chinnery and North, 1975; or those used in Bath, 1978 and Anderson, 1979. From all these models, the one published by Gutenberg and Richter, 1944 (GR) modified by Cosentino et al., 1977, is the most

widely used model for characterizing the source. According to Gutenberg-Richter modified (5), this model (Eq. (5)) provides a relationship between the cumulative rate for different magnitudes $\dot{N}(m)$, the rate of earthquakes being the number of seisms of magnitude less than an established minimum (*Nmin*), generated at a certain time and in a particular area.

$$\dot{N}(m) = Nmin \cdot \left[ \frac{e^{-\beta(m)} - e^{-\beta(Mmax)}}{(e^{-\beta(Mmin)} - e^{-\beta(Mmax)})} \right] \tag{5}$$

From the rate of seismic moment, it can be established a relationship between this parameter and a recurrence model type

GR through the expression of Anderson, 1979, Eq. (6).

$$\dot{M}_0 = \int_{Mmin}^{Mmax} \dot{n}(m) \cdot Mo(m) dm \tag{6}$$

where, $\dot{M}o$ is the seismic moment rate, $\dot{n}(m)$ is the rate of magnitude *m* earthquakes and *Mo(m)* is the seismic moment generated in an earthquake of magnitude *m*.

Moreover, Anderson and Lucco (1985) propose relationships between the rate of cumulative seismic moment $\dot{M}o$ and three

models of recurrence: Gutenberg-Richter truncated, Gutenberg-Richter modified, and the recurrence model proposed by Main and Burton, 1981. In this paper, the model GR-modified is used, with the Eq. (7), where the cumulative rate of earthquakes of magnitude minimum *Nmin* is dependent on moment rate among other parameters, this expression is derived from the Eq. (5) and Eq. (6).

$$Nmin = \frac{\dot{M}_0(\bar{d} - \beta)\left(e^{-\beta\, M\, min} - e^{-\beta\, M\, max}\right)}{\beta\left[e^{-\beta\, M\, max}\, M_0(M_{max}) - e^{-\beta\, M\, min}\, M_0(M_{min})\right]} \tag{7}$$

where *Nmin* is the cumulative rate of earthquakes of magnitude greater than or equal to *Mmin* ; β is a parameter that defines the proportion of earthquakes as a function of their magnitude and will be set at 1.84-2.76 ; *Mmax*, *Mmin*, maximum and




minimum magnitude of truncation; *Mo(Mmax)* and *Mo(Mmin)*, the seismic moment (co - seismic) that would be released in these possible maximum and minimum earthquake, respectively, obtained from the expression of Hanks and Kanamori, 1979 and $\bar{d}=1.5\cdot ln\,(10)$.

This expression allows to deduce the rate of seismic activity of the fault truncated at a maximum and minimum size, with the rate of seismic moment of the fault.

In Fig. 7, the earthquake cumulative rate is presented for different magnitudes, and for each of the segments of the reverse faults in Quito. From this figure, Table 3 shows the recurrence periods for different magnitude ranges.

As elastic response spectra for maximum magnitudes in each fault segment was obtained, it is interesting to observe the last row of Table 3. For the Puengasí fault, an earthquake of magnitude 6.4 is expected in a time interval between 1224 and 2190 years ; for the Ilumbisí-La Bota fault, an earthquake of magnitude 6.2 is expected between 610 and 981 years.

However, it is known that an earthquake of magnitude 6.4 in 1587 in northern Quito, associated with blind faults, caused great damage in the city (Beauval et al., 2010) ; since then there has not been an earthquake of magnitude greater than 6 associated with these failures, which suggests that there is a significant accumulation of energy in these faults.

Moreover, the date on which a magnitude greater than 6.0 earthquake associated with the Puengasí or Ilumbisí – La Bota faults occurred is not known, so that compliance with the period of recurrence in these failures can be in few or many years. The truth is that more than four centuries are gone without a strong earthquake and so the likelihood of a severe earthquake in these two fault segments increases

## 6 Commentary and conclusions

Quito is on or very close to five segments of reverse faults called from south to north: Puengasí, Ilumbisí-La Bota, Carcelen-El Inca, Bellavista-Catequilla and Tanhaguilla. These segments are active so that in this paper monitoring spectra for the horizontal movement of the floor were computed, and for the five areas of the city: South, South-Central, Central, North and North Central; according to the classification of soils of the NEC-15.

These spectra were obtained using the models of Abrahamson et al., 2013; Campbell and Borzognia, 2013 and Chiou and Youngs, 2013; and considering the maximum amount of energy that could be generated in each segment of the blind faults; spectra for 50 and 84% confidence level are also found, finding that the latter have spectral ordinates higher than those found with the NEC-15, so it is important that new construction projects to be built in Quito consider the spectra found in this paper.

Moreover, the recurrence periods of each of these faults was determined by applying the modified and truncated Gutenberg and Richter model, and finding that the maximum magnitude expected in each segment, with which the control spectra were found, is between 549 and 2190 years.



Finally, the structures should be designed for two spectrums denominated: design and maximum considered spectra. The design spectrum is similar to that stipulated in the NEC-15 and the maximum considered spectrum is that studied in this paper with a confidence of 84%.

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





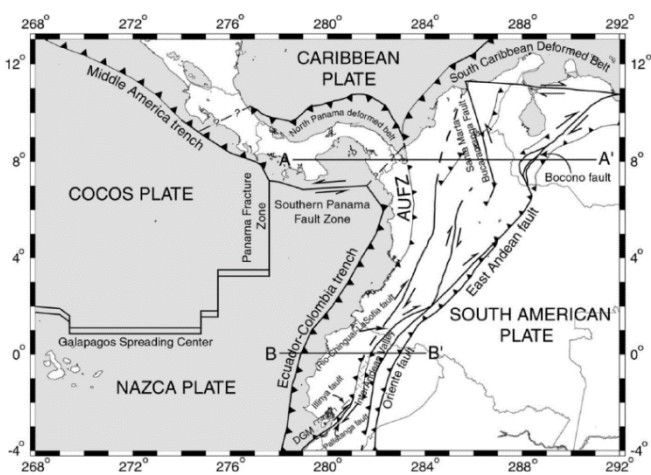

**Figure 1 Plate tectonics in the nortwest of South America. (Trenkamp et al. 2002)**


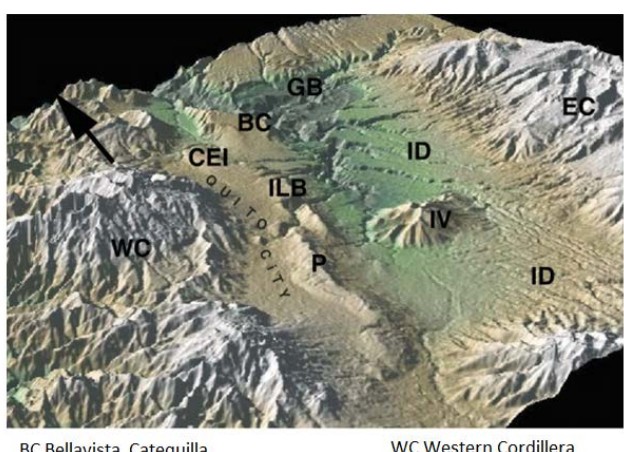

| | |
|---|---|
| BC Bellavista  Catequilla | WC Western Cordillera |
| ID Interandean Depression | WC Eastern Cordillera |
| IV Ilaló Volcano | P Puengasí |
| GB Guayllabamba Basin | ILB Ilumbisí - La Bota |
| | CEI Carcelén - El Inca |

**Figure 2  3D view of the Andean valleys. (Alvarado et al. 2014)**

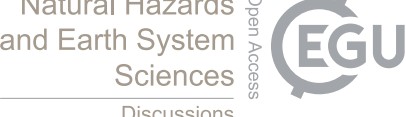



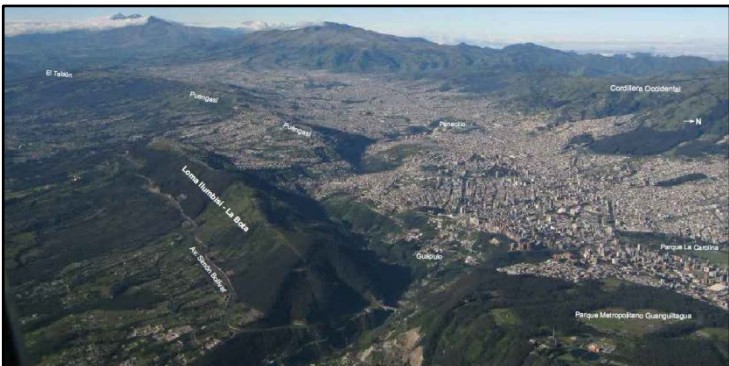

**Figure 3 The Ilumbisi-La Bota hill at the front, and the Puengasí hill at the rear**





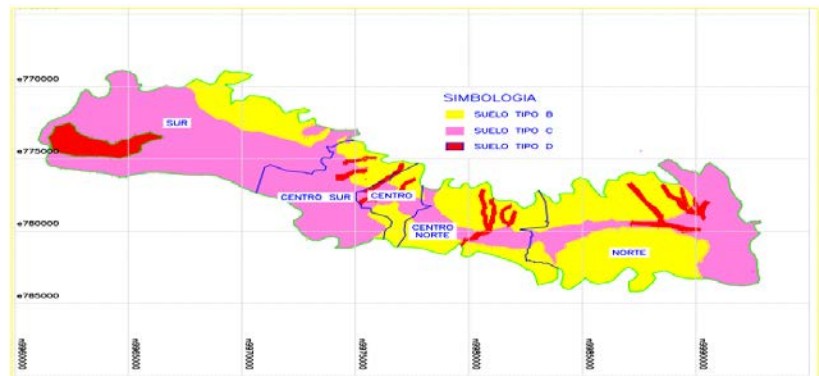

**Figure 4 Soil classification in Quito.**

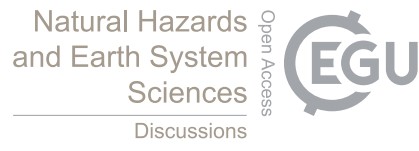

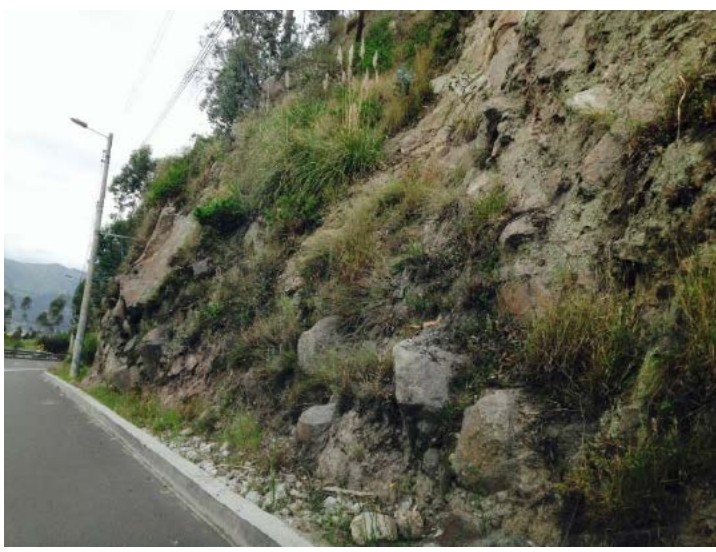

**Figure 5 Rocky soil in Quito, Vs30= 466.27 m/s. Metropolitan Park.**





**Figure 6  Spectra for five zones of Distrito Metropolitano de Quito**



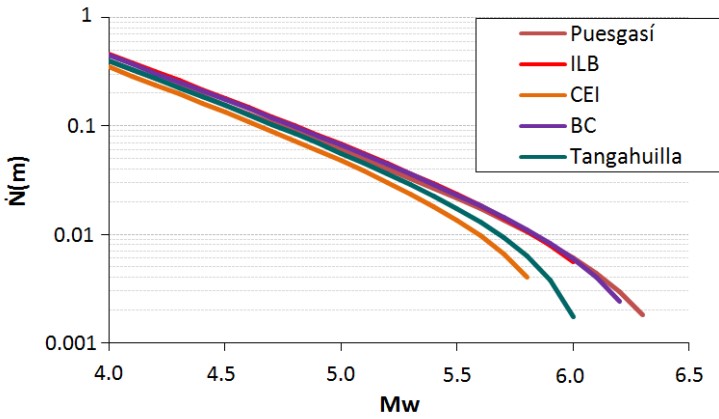

**Figure 7 Earthquake cumulative rate for different magnitudes Ṅ(m). (Rivas-Medina et al. 2014)**

20



| Segmento | Area of ruptura (km)$^2$ | Length of ruptura (km) | Magnitude (Mw) |
|---|---|---|---|
| Puengasí | 259 | 22 | 6.4 |
| ILB | 176 | 15 | 6.2 |
| CEI | 82 | 7 | 5.9 |
| BC | 191 | 17.5 | 6.3 |
| Tangahuilla | 108 | 12 | 6.0 |

**Table 1 Area and length of rupture of the fault segments and maximum expected magnitude (Alvarado et al. 2014)**








| Model | ASK13 | CB13 | CY13 |
|---|---|---|---|
| **Source parameter** | | | |
| Magnitude | X | X | X |
| Mechanisms | X | X | X |
| Dip | X(HW) | X | X |
| Rupture width (W) | X | X | X |
| Hanging-Wall effect (HW) | X | X | X |
| Depth of the rupture surface ($Z_{TOR}$) | X | X(HW) | X |
| Depth of the hypocenter($Z_{hyp}$) | | X | |
| **Types of distances** | | | |
| $R_X$ distance | X(HW) | X(HW) | X(HW) |
| Joyner-Boore distance ($R_{jb}$) | X(HW) | X(HW) | |
| Rupture surface distance ($R_{rup}$) | X | X | X |
| $R_Y$ distance | X(HW) | | |
| $R_{yo}$ distance | X(HW) | | |
| **Site effects** | | | |
| $V_{s30}$ | X | X | X |
| $Z_{1.0}$ | X | | X |
| $Z_{2.5}$ | | X | |
| **Other effects** | | | |
| Directivity | | | X |
| Attenuation | X | X | X |

**Table 2 Parameters considered by the selected strong movement models**





| Magnitude ranges | Recurrence periods (years) | | | | |
|---|---|---|---|---|---|
| | **PUENGASÍ** | **ILB** | **CEI** | **BC** | **Tangahuilla** |
| **[5.0 – 5.5)** | 20 - 35 | 18 - 30 | 27 - 39 | 18 - 31 | 23 - 34 |
| **[5.5 – 6.0)** | 62 - 87 | 56 - 75 | 85 - 130 | 58 - 78 | 65 - 97 |
| **[6.0 <** | 164 - 262 | 179 - 279 | | 169 - 279 | 579 - 1016 |
| **Mmax** | 1224 - 2190 *(Mw6.4)* | 610 - 981 *(Mw6.2)* | 549 - 952 *(Mw5.9)* | 908 - 1630 *(Mw6.3)* | 579 - 1016 *(Mw6.0)* |

**Table 3. Recurrence periods, Gutenberg and Richter modified model.**