# Peer review of "Control Spectra for Quito"

_Natural Hazards and Earth System Sciences, 2016_

## Referee Comment (RC1) · F. Moreu (Referee) · 12 Jul 2016

In this manuscript, the authors present earthquake spectra for the city of Quito which is located at the top of a very earthquake-prone fault system. Moreover, the recurrence period of earthquakes are also delivered. The spectra provided here are the results of a study aggregating a variety of sources such as multiple ground motion databases, local soil profile studies and fault characteristics. The work presented here has a potential to be used as a new design spectra in designing buildings or understanding the current state of the existing structure. Although the research is innovative and a useful tool has clearly been developed, this reviewer feels that the authors did not reflect the use of the method(s) here to its entire potential. The English is from time to time imprecise, and can be improved in a new version of the paper. There are some problems with data presented within the figures. Reviewer thinks however that this manuscript still has a big potential. Therefore, this manuscript is recommended for publication with

minor revisions. Below are some suggestions for the authors to consider, in addition to any suggestions offered by the editor.

Pg1. Line 9: In the abstract, please avoid using references unless this work is a major extension of the referenced paper. Pg1. Line 10: Reviewer suggests an improvement in the choice of words for the fluency. For instance . . ., which are determined by utilizing some of the . . . Pg1. Line 11: Avoid references. Pg1. Line 12: Reviewer doesn't understand what authors are trying to quantify when they say maximum amount. Maximum amount of what? Peak acceleration? Pg1. Line 13: It is unclear from abstract that if each location has different soil type. Please clarify. Pg1. Line 14: No need for abbreviation here (NEC-15) Pg1. Line 16: Reviewer suggests style improvement. If possible, the sentence should be divided into two sentences. Pg1. Line 17: Please remove the references. Pg1. Line 23-24: Reviewer suggest a simplification such as: . . .with an average slip rate of 3.0- 4.5 mm per year. . . Pg1. Line 24: There is a dot before and after the reference. Pg1. Line 26: Reviewer failed to make connection between non-uniform movements of the value with the turnover rate of 2-4mm of Quito's fault system. Pg1. Line 26: What is the SFQ abbreviation for? For the abbreviations used for the first time, please use the full name. Pg1. Line 28-29: Why is the reference not at the end of the sentence? Pg1. Line 29: Please improve the style: For example: . . . Since then, there has not been an earthquake with a magnitude larger than 6.0... Introduction general comments: Reviewer does not understand the motivation of the paper clearly. Could authors explicitly mention and quantify scientifically why current spectra are poor? Pg2. Line 11: Reviewer is not sure what author is implying here: . . .Being "similar" the maximum expected magnitude in each fault segment. . . Is there a punctuation missing? Pg2. Line 20: Please use punctuations to increase readability: . . .August 12, 2014, which had. . . Pg2. Line 24: Author thinks that this sentence is missing a verb. . . .Quito is 2800 m above the sea level and the valleys are 2400 m . . . Fig 3: Reviewer strongly suggests that authors should use higher quality picture with more readable text on it. Pg3. Line 1: Please provide a reference for the refraction study. Fig 3: Typo in the table headings: ruptura rupture Pg3. Line 4: Reviewer

thinks that this sentence is missing some parts: . . .obtained using equations provided by Leonard, 2010. . . Pg3. Line 6: No need for 'as'. Pg3. Line 6-7: Reviewer suggest to break this sentence into two sentences for the readability. Pg3. Line 7-9: Reviewer is lost here. Are authors referring to table 2 here? Also what do authors mean when they say 'it is worked as an increment?' Also if Z_TOR value is going to be described after the equation, what is the point of using this variable? Pg3. Line 14-15: Are those equations giving average value or observed value? If it is average value, why are they not called E[Z_TOR]_1 and E[Z_TOR]_2? If both of the equations are calculating the depth, they could be named as (2.a.) and (2.b.). Please revise the equations. Otherwise, it is somewhat confusing for the readers. Pg3. Line 16: Why is unit of Z_TOR not mentioned before? Pg3. Line 16-17: Reviewer is having difficulty understanding what the claim is. Clarification is needed. Pg3. Line 21: Reviewer is having difficulty understanding what $\varepsilon$ is. It would help if the confidence level formula was presented. Pg3. Line 23: Reviewer suggests not the use the word, 'this issue'. But authors have the freedom to leave as is. Pg3. Line 32: Reviewer suggests the following modification. . . .the model of CY13 employed with. . . Pg4. Line 1-2: Reviewer is having difficulty understanding this sentence. Pg4. Line 9: Reviewer suggest to use comma: . . . in the second model, . . . Pg4. Line 9: What does DPP stands for? Pg4. Line 11: There is an extra space before in addition. Pg4. Line 11-14: Reviewer suggests the following change: . . . as a parameter in determining the spectral acceleration, which is not ignored in the other models. . . Pg4. Line 11-14: There are too many paragraphs in this section, one paragraph per sentence. Fig. 4: A higher quality picture should be used. Likewise, legends should be in English. Pg4. Line 16-17: Author should divide this sentence. Pg4. Line 21-22: Reference required. Is there a study authors can point to regarding the magnitude of the earthquake? Pg4. Line 22-23: When authors mention the distance, are they referring to the depth, rupture length of the distance of the fault from the city? Pg4. Line 24-26: Reviewer suggests the following change. . . .There have been several studies to classify soils in Quito. . . Also, authors should use the punctuations correctly. In addition, reviewer thinks that authors should use transition

words instead of generous use of punctuations. Pg4. Line 27-28: Reviewer thinks using semi-column does not replace the linking words and suggests the sentence should be either broken down into two sentences or should be connected with a linking word. Another idea is enumeration of each sentences. Pg4. Line 29-30: Are those all of the soil types in CEC-2000? Reviewer is confused. Pg4. Line 31: Why did authors decide to divide the sentences with comma? The general style is somewhat inconsistent. Inconsistencies in the format makes the scientific message difficult to come across. Pg4 Line 27 – Pg5 Line 3: Reviewer didn't comprehend why it is important for authors to mention all the soil types in old and new code. Authors failed to describe the relevant information to Fig. 4. If only 2015 code are relevant to the soil types, why are authors mentioning the old code? What is the significance of this information? Pg5. Line 4-6: Why is this study conducted? What is the purpose of the refraction study? And most importantly why do authors mention this here? If they want to verify that the soil type given in the map aligns with the study results, shouldn't be there multiple research studies conducted at multiple points verifying the soil conditions. Reviewer feels lost in this part of the paper. Pg5. Line 5: Is it floor type or soil type or soil profile? Readers expect authors to use consistent terminology. Pg5. Line 7-8: How is it relevant that soil types have good compressive strength to the paper? Authors should clarify why the information they presented is important when they are stating some facts. Pg5. Line 10: Reviewer is not sure what authors are claiming in the first sentence. What kind of meshes are formed. What is meshed? Why is it meshed? Pg5. Line 14-20: What is the methodology used to obtain this information? Fig. 6: This figure is a combination of 15 figures. When printed, all details are lost. Authors should manage the plots well, if they want to share their results and to be understood properly. The legends are unreadable. Increasing the quality of the figures will increase the quality of the paper significantly. Pg5. Line 25: Reviewer is curious if there is a word missing here: . . ..first and modeled . . .. Pg5. Line 26: When authors mention 'this parameter', are they referring to the recurrence interval, seismic potential (if there is such a parameter) or seismic moment? Pg6. Line 1: There is an extra space between (Table 1) and comma. Authors

should delete the space. Pg6. Line 1 and Line 1.2: Reviewer thinks that the number of these subsections are wrong. Pg6. Eq 4: Although the equation is shown, no other info provided on fault plane area or shear modulus. Is it same as rupture area? Are authors expecting readers to guess their assumptions for replicating the work? Pg6. Line 15: There is a (5) after Gutenberg-Richter modified. Is it a typo? Pg6. Line 15: Reviewer thinks that there is a word missing: Gutenberg-Richter modified (model?) Pg6. Eq 5: Reviewer is curious about "e" number. Is this a exponential relationship or "e" is another number? Pg6. Line 19: Reviewer suggest the following change: . . . moment, a relationship can be established . . . If the initial claim was that the spectra are out-of-date, why didn't authors compare the new control spectra to the old ones and showed us how different the spectra are? Pg6. Line 21: Mo is already explained. Why is repeated again? If it is not explained before, is this Mo different from the Mo in eq 4? Pg6. Line 26-28: Reviewer thinks that this sentence contains two different sentences. Pg6. Line 31: Why is beta set to 1.84-2.76? Pg6. Line 30-31: There are spaces before semi-column. Please delete them. Reviewer fails to understand how the occurrence is important deriving the control spectra. Authors should explicitly discuss why the rate of seismic activity is discussed within this paper. Were they not calculated before? Do authors propose a novel method to calculate it? Readers want to understand the significance of this research study. Pg7. Line 18: There are two sections which have the same number '6'. Authors should be careful when enumerating the sections. Pg7. Line 20-22: The style can be changed. Please improve the wording. Reviewer doesn't grasp the meaning of this sentence very well. Pg7. Line 23-27: Reviewer thinks the sentence is long enough that reader may lose the focus. It is hard to understand the meaning. Pg7. Line 28: Grammatical error. Plural subject – singular verb form Pg7. Line 28: Reviewer is not sure when authors say 'applying the models'. Do the authors applied these methods on some data sets, or did they just use the methods? Pg7. Line 28-30: This sentence should be broken down into two sentences. Pg8. Line 1-3: Authors should improve wording for the readability.

---

## Short Comment (SC1) · 14 Jul 2016

Thanks for your comments

we will consider all to present a new version of the paper to the editor

---

## Author Comment (AC1) · 27 Jul 2016

**Control Spectra for Quito**

Roberto Aguiar[1], Alicia Rivas-Medina[1,2], Pablo Caiza[1], Diego Quizanga[3]

[1] Departamento de Ciencias de la Tierra y la Construcción, Universidad de Fuerzas Amadas ESPE, Quito, Ecuador

[2] Departamento de Cs. Geodésicas y Geomática, Universidad de Concepción - Campus Los Ángeles, Los Ángeles, Chile.

5  [3] Departamento de Ingeniería Civil, Escuela Politécnica Nacional, Quito, Ecuador

*Correspondence to*: Roberto Aguiar (rraguiar@espe.edu.ec)

**Abstract.** The Metropolitan District of Quito is divided into five areas: south, south-central, central, north-central and north. It is located on or very close to segments of reverse blind faults: Puengasí, Ilumbisí-La Bota, Carcelen-El Inca, Bellavista-Catequilla and Tangahuilla, making it one of the most seismically dangerous cities in the world. For each of the urban areas

10  of Quito, elastic response spectra are presented in this paper, which are determined by utilizing some of the new models of the PEER's NGA-West2 Program. These spectra are calculated considering the maximum magnitude that could be generated by the rupture of each fault segments, and taking into account the soil type that exists in different point of the city according to the Norma Ecuatoriana de la Construcción, 2015. Subsequently, the recurrence period of earthquakes of high magnitude in each fault segment is determined from the physical parameters of the fault segments (size of the fault plane and slip rate) and

15  the pattern of recurrence of type Gutenberg-Richter earthquakes with double truncation magnitude (Mmin and Mmax) 
[revised manuscript text omitted]
{\exp\left(-\beta(m)\right) - \exp\left(-\beta(Mmax)\right)}{\exp\left(-\beta(Mmin)\right) - \exp\left(-\beta(Mmax)\right)} \right] \tag{4}$$

From the rate of seismic moment, it can be established a relationship between this parameter and a recurrence model type GR through the expression of Anderson, 1979, Eq. (5).

10  $$\dot{M}_0 = \int_{Mmin}^{Mmax} \dot{n}(m) \cdot Mo(m)dm \tag{5}$$

Where $\dot{n}(m)$ is the rate of magnitude $m$ earthquakes and $Mo(m)$ is the seismic moment generated in an earthquake of magnitude $m$.

Moreover, Anderson and Lucco (1985) propose relationships between the rate of cumulative seismic moment $\dot{Mo}$ and three models of recurrence: Gutenberg-Richter truncated, Gutenberg-Richter modified, and the recurrence model proposed by
15  Main and Burton, 1981. In this paper, the model GR-modified is used, with the Eq. (6), where the cumulative rate of earthquakes of magnitude minimum $Nmin$ is dependent on moment rate among other parameters, this expression is derived from the Eq. (4) and Eq. (5).

$$Nmin = \frac{\dot{M}_0(\bar{d}-\beta)\left(\exp\left(-\beta(Mmin)\right) - \exp\left(-\beta(Mmax)\right)\right)}{\beta \left[\exp\left(-\beta(Mmax)\right)M_0(M_{max}) - \exp\left(-\beta(Mmin)\right)
[revised manuscript text omitted]

[Figure]

BC Bellavista  Catequilla
ID Interandean Depression
IV Ilaló Volcano
GB Guayllabamba Basin

WC Western Cordillera
WC Eastern Cordillera
P Puengasí
ILB Ilumbisí - La Bota
CEI Carcelén - El Inca

**Figure 2  3D view of the Andean valleys. (Alvarado et al. 2014)**

[Figure]

**Figure 3 The Ilumbisi-La Bota hill at the front, and the Puengasí hill at the rear**

[Figure]

**Figure 4 Soil classification in Quito.**

[Figure]

**Figure 5 Rocky soil in Quito, Vs30= 466.27 m/s. Metropolitan Park.**

[Figure]

**Figure 6  Spectra for five zones of Distrito Metropolitano de Quito**

[Figure]

**Figure 7 Earthquake cumulative rate for different magnitudes Ṅ(m) (Rivas-Medina et al. 2014).**

20

| Segment | Area of rupture (km)$^2$ | Length of rupture (km) | Magnitude (Mw) |
|---|---|---|---|
| Puengasí | 259 | 22 | 6.4 |
| ILB | 176 | 15 | 6.2 |
| CEI | 82 | 7 | 5.9 |
| BC | 191 | 17.5 | 6.3 |
| Tangahuilla | 108 | 12 | 6.0 |

**Table 1 Area and length of rupture of the fault segments and maximum expected magnitude (Alvarado et al. 2014).**

20

25

30

35

40

| Model | ASK13 | CB13 | CY13 |
|---|---|---|---|
| **Source parameter** | | | |
| Magnitude | X | X | X |
| Mechanisms | X | X | X |
| Dip | X(HW) | X | X |
| Rupture width (W) | X | X | X |
| Hanging-Wall effect (HW) | X | X | X |
| Depth of the rupture surface ($Z_{TOR}$) | X | X(HW) | X |
| Depth of the hypocenter($Z_{hyp}$) | | X | |
| **Types of distances** | | | |
| $R_X$ distance | X(HW) | X(HW) | X(HW) |
| Joyner-Boore distance ($R_{jb}$) | X(HW) | X(HW) | |
| Rupture surface distance ($R_{rup)}$ | X | X | X |
| $R_Y$ distance | X(HW) | | |
| $R_{yo}$ distance | X(HW) | | |
| **Site effects** | | | |
| $V_{s30}$ | X | X | X |
| $Z_{1.0}$ | X | | X |
| $Z_{2.5}$ | | X | |
| **Other effects** | | | |
| Directivity | | | X |
| Attenuation | X | X | X |

**Table 2 Parameters considered by the selected strong movement models**

| Magnitude ranges | Recurrence periods (years) | | | | |
|---|---|---|---|---|---|
| | PUENGASÍ | ILB | CEI | BC | Tangahuilla |
| [5.0 – 5.5) | 20 - 35 | 18 - 30 | 27 - 39 | 18 - 31 | 23 - 34 |
| [5.5 – 6.0) | 62 - 87 | 56 - 75 | 85 - 130 | 58 - 78 | 65 - 97 |
| [6.0 < | 164 - 262 | 179 - 279 | | 169 - 279 | 579 - 1016 |
| Mmax | 1224 - 2190 *(Mw6.4)* | 610 - 981 *(Mw6.2)* | 549 - 952 *(Mw5.9)* | 908 - 1630 *(Mw6.3)* | 579 - 1016 *(Mw6.0)* |

**Table 3. Recurrence periods, Gutenberg and Richter modified model.**

---

## Referee Comment (RC2) · Anonymous Referee #2 · 22 Nov 2016

22 November 2016

This is overall a well organized paper, but needing work in places to bring it up in its formatting and how references are used.

Below are my comments in no particular order of importance.

(a) Abstract. Avoid references in the abstract. (b) Abstract. This reads more like an introduction to a paper, not a substantive summary. Review what it means to write an abstract. (c) Units. Rather than mm per year, it should be mm yrˆ-1 (where ˆ indicates superscript). (d) In-text citations. Throughout, you cannot have a "." before your in-text citations. They have to be part of the sentence where they occur. (e) Make sure that ALL facts and information have proper citations. I see a number of cases where you give facts/information, but we do not know how you know this information. So for example (there are many) "Obviously this movement is not uniform along the fault, so

that the blind reverse faults system of Quito, SFQ, which is part of the megafault, has a turnover rate of 2-4 mm per year." or "It is also well known that local soil conditions are a key factor affecting the spectrum form." We don't know how you know this information or what the references are for either of these statements. You need to go through the entire manuscript and ensure all facts and information are clear. (f) In the introduction (or another section) please give a bit more background about others that have done control spectra, so we have a better idea of their limitations, strengths, and history. (g) Please provide early on, a table that includes all variables (and units), and refer to that table. (h) Figure 6. This is the key to the paper, but VERY hard to see. Consider whether parts of this also need to be uploaded as suplementary figure (e.g., as high resolution, or the data). (i) Figure 7. This does not come out well. Please use different weights for lines, and dashes for some, dash dot for other. Colour alone is not enough to distinguish the lines. (j) Please convert all spanish words to English. (k) Overall, and particularly in the final results, please provide a discussion of uncertainties (of the data, of the method, and particularly the results).

―――――――――――――――――――――